# Exploring Weak-to-Strong Generalization for CLIP-based Classification

**Jinhao Li**                                                          *jinhao.li2@unimelb.edu.au*
*School of Computing and Information Systems*
*University of Melbournem, Australia.*

**Sarah M. Erfani**                                                   *sarah.erfani@unimelb.edu.au*
*School of Computing and Information Systems*
*University of Melbourne, Australia.*

**Lei Feng**                                                                    *fenglei@seu.edu.cn*
*School of Computer Science and Engineering*
*Southeast University, China.*

**James Bailey**                                                          *baileyj@unimelb.edu.au*
*School of Computing and Information Systems*
*University of Melbourne, Australia.*

**Feng Liu**                                                              *feng.liu1@unimelb.edu.au*
*School of Computing and Information Systems*
*University of Melbourne, Australia.*

**Reviewed on OpenReview:** *https://openreview.net/forum?id=quE8gDDegf*

## Abstract

Aligning large-scale commercial models with user intent is crucial to preventing harmful outputs. Current methods rely on human supervision but become impractical as model complexity increases. When models surpass human knowledge, providing accurate feedback becomes challenging and inefficient. A novel solution proposed recently is using a weaker model to supervise a stronger model. This concept leverages the ability of weaker models to perform evaluations, thereby reducing the workload on human supervisors. Previous work has shown the effectiveness of weak-to-strong generalization in the context of language-only models. Extending this concept to vision-language models leverages these insights, adapting the proven benefits to a multi-modal context. In our study, we explore weak-to-strong generalization for CLIP-based classification. We propose a method, *class prototype learning* (CPL), which aims to enhance the classification capabilities of the CLIP model, by learning more representative prototypes for each category. Our findings indicate that, despite using a simple loss function under weak supervision, CPL yields robust improvements in targeted scenarios, particularly when pretraining is limited. Extensive experiments demonstrate that our approach is effective under these settings, achieving a 3.67% improvement over strong baseline methods.

## 1 Introduction

Large language models (LLMs), such as GPT 4o (Achiam et al., 2023), Claude 3 (Anthropic, 2024) and Gemini 1.5 (Reid et al., 2024), have made significant strides in enhancing performance across a spectrum of natural language processing tasks. However, despite their successes, ensuring that these models align with human expectations and intentions remains a formidable challenge (Burns et al., 2023). Increasing the size of language models does not necessarily improve their ability to follow user intent, as they can still produce

untruthful, toxic, or unhelpful outputs, indicating a lack of alignment with their users (Ouyang et al., 2022). Alignment with user intent is crucial for deploying these models effectively in practice (Bai et al., 2022). Traditional alignment techniques often rely heavily on human supervision (Christiano et al., 2017; Stiennon et al., 2020; Ouyang et al., 2022; Glaese et al., 2022; Bai et al., 2022), requiring evaluators to provide feedback on model outputs. However, as the complexity and intricacy of model outputs increase, the feasibility and scalability of this approach diminishes (Burns et al., 2023). As a result, there is a need to effectively align LLMs with human values without overly burdening human evaluators.

A recent study (Burns et al., 2023) explored a novel approach known as weak-to-strong generalization to address the challenge of aligning strong models with human feedback. This strategy leverages a weaker model to supervise a more robust one, presenting a promising method to enhance model alignment. The study by Burns et al. (2023) demonstrates the effectiveness of this weak-to-strong learning approach, where finetuning strong models with knowledge generated by their weaker counterparts consistently improves performance. For example, in natural language processing tasks, finetuning GPT-4 with supervision from a GPT-2-level model significantly enhances GPT-4's performance. This approach highlights the viability of weak-to-strong learning as a solution for better model alignment, demonstrating that even weaker models can provide valuable guidance for improving stronger models. While the technique proves effective, applying it to Vision-Language Models (VLMs) is far from straightforward. VLMs face unique challenges in aligning complex multimodal tasks, making it essential to thoroughly explore the method's applicability and limitations in this context. Unlike in natural language tasks, where text-based guidance can be more straightforward, VLMs must align both visual and textual information, making supervision from weaker models significantly more challenging. The complexity of managing two distinct modalities introduces difficulties in ensuring coherent feedback across image and text domains, necessitating a more nuanced approach when adapting weak-to-strong generalization to VLMs. Our goal is to rigorously investigate the weak-to-strong paradigm within VLMs, as this problem extends beyond a mere adaptation of previous work.

In this study, we explore weak-to-strong generalization for CLIP-based classification, recognizing it as a crucial starting point in VLMs. Existing VLMs (Radford et al., 2021; Jia et al., 2021) take classification as the basic task to evaluate the alignment of images and texts. Also in our task setting, classification makes it easier for us to design simulation experiments and establish a benchmark. In this context, we introduce a method called *class prototype learning* (CPL). CPL involves generating class prototypes that encapsulate the characteristics of each class using weak supervision. This method effectively mitigates the false signals typically generated by weak supervision, thereby showcasing superior performance. Moreover, when compared to conventional methods for adapting VLMs to downstream tasks, such as prompt tuning (Zhou et al., 2022b; Jia et al., 2022), our CPL approach proves to be more efficient. This efficiency arises from the fact that CPL eliminates the need to employ a text encoder during the fine-tuning phase. By streamlining the adaptation process, CPL offers a more resource-effective solution while maintaining high-performance levels.

We conduct extensive experiments to evaluate the performance of the proposed method, CPL, using the DomainNet dataset (Peng et al., 2019), which includes six diverse visual domains. The dataset is divided into training and test sets, and the experiment involves training weak models, generating weak supervision sets, fine-tuning strong models with weak supervision, and comparing the results to strong model training with ground truth labels. Various baselines are used for comparison. Our results show that the CPL achieves the highest average accuracy across all domains, significantly outperforming other methods. In particular, CPL shows substantial improvements for challenging domains like Infograph and handles domain-specific features effectively, despite CLIP's lower zero-shot performance in domains like QuickDraw. This illustrates the robustness and effectiveness of CPL in weak-to-strong generalization scenarios.

We summarize the main contributions of our work:

(i) **Exploring weak-to-strong generalization for CLIP-based classification**: Previous work (Burns et al., 2023; Guo et al., 2024) has shown the effectiveness of weak-to-strong generalization in LLMs. Extending this concept to VLMs leverages these insights, adapting the proven benefits to a multi-modal context.

(ii) **Proposing CPL**: We present a method that effectively leverages class prototype representations through weak supervision to enhance the classification performance of VLMs, such as CLIP (Radford et al., 2021).

(iii) **Conducting simulation experiments**: We design a simulation experiment within the VLMs framework based on DomainNet (Peng et al., 2019) to study this problem, and establish a benchmark in this context. Our experiment resulted in a 3.67% improvement over baseline methods.

## 2    Related Work

**Vision-language models.**    VLMs integrate visual and textual information, enabling a multifaceted understanding and interaction with multimodal content. CLIP (Radford et al., 2021) exemplifies this approach, leveraging contrastive learning to align images with textual descriptions effectively. This model demonstrates robust zero-shot capabilities, where it can recognize images or concepts it was not explicitly trained on. The effectiveness of CLIP and similar models, such as ALIGN (Jia et al., 2021), Flamingo (Alayrac et al., 2022), BLIP (Li et al., 2022) and Llava (Liu et al., 2023), arises from their ability to generalize from vast amounts of web-collected data, learning nuanced, multimodal representations that are applicable across various tasks and domains.

**Vision-language prompt tuning.**    Research has also focused on improving prompt-based learning and fine-tuning methods, such as CoOp (Zhou et al., 2022b) and CoCoOp (Zhou et al., 2022a), which adapt VLMs more effectively to specific tasks by learning customized prompt strategies. CoOp transforms static text prompts into dynamic, learnable components. This allows prompts to adjust during training, aligning model responses with task-specific needs, and improving performance, especially in zero-shot or few-shot settings. Following CoOp, several studies Lu et al. (2022); Sun et al. (2022); Derakhshani et al. (2023); Zhu et al. (2023); Gao et al. (2024) have advanced prompt tuning to enhance model performance.

**Knowledge distillation.**    Knowledge distillation (Hinton et al., 2015; Ahn et al., 2019; Zhao et al., 2022; Jin et al., 2023) is an effective model compression technique in which a smaller, more efficient student model learns from a larger, more complex teacher model. The conventional method for knowledge distillation involves training the student model to minimize the difference between its predicted probability distribution and that of the teacher model, often measured using Kullback-Leibler (KL) divergence. However, weak-to-strong generalization offers an alternative by having strong models supervised by weaker models.

**Weak-to-strong generalization.**    The concept of weak-to-strong generalization, initially introduced by Burns et al. (2023), presents a promising approach for aligning super-intelligent models with human values. This study emphasizes the significance of the issue and provides experimental evidence to support its feasibility. Building on this framework, Guo et al. (2024) introduces a dynamically adjusted confidence loss and demonstrates the effectiveness of their method in the context of visual foundation models. Therefore, based on those previous work, we explore the weak-to-strong generalization for VLMs.

## 3    Preliminaries

In this section, we outline the preliminary studies considered in this paper.

**CLIP-like vision-language models.**    The CLIP model (Radford et al., 2021) employs a vision encoder $f_{\text{vision}}^{\text{s}}$ and a text encoder $f_{\text{text}}^{\text{s}}$, which jointly learn to map visual inputs $x_i$ and textual inputs $t_j$ into feature embeddings $r_i = f_{\text{vision}}^{\text{s}}(x_i)$ and $r_j = f_{\text{text}}^{\text{s}}(t_j)$, respectively. These embeddings are projected into a shared latent space where their similarity is measured by cosine similarity, $\cos(r_i, r_j)$. By maximizing the similarity of positive pairs $(r_i, r_j)$ and minimizing the similarity of negative pairs sampled from the dataset, CLIP optimizes the contrastive loss function:

$$\mathcal{L}_{\text{CLIP}} = \frac{1}{N} \sum_{n=1}^{N} \log \frac{\exp(\cos(r_{i_n}, r_{j_n})/\tau)}{\sum_{k=1}^{N} \exp(\cos(r_{i_n}, r_{j_k})/\tau)} + \frac{1}{N} \sum_{n=1}^{N} \log \frac{\exp(\cos(r_{j_n}, r_{i_n})/\tau)}{\sum_{k=1}^{N} \exp(\cos(r_{j_n}, r_{i_k})/\tau)},$$

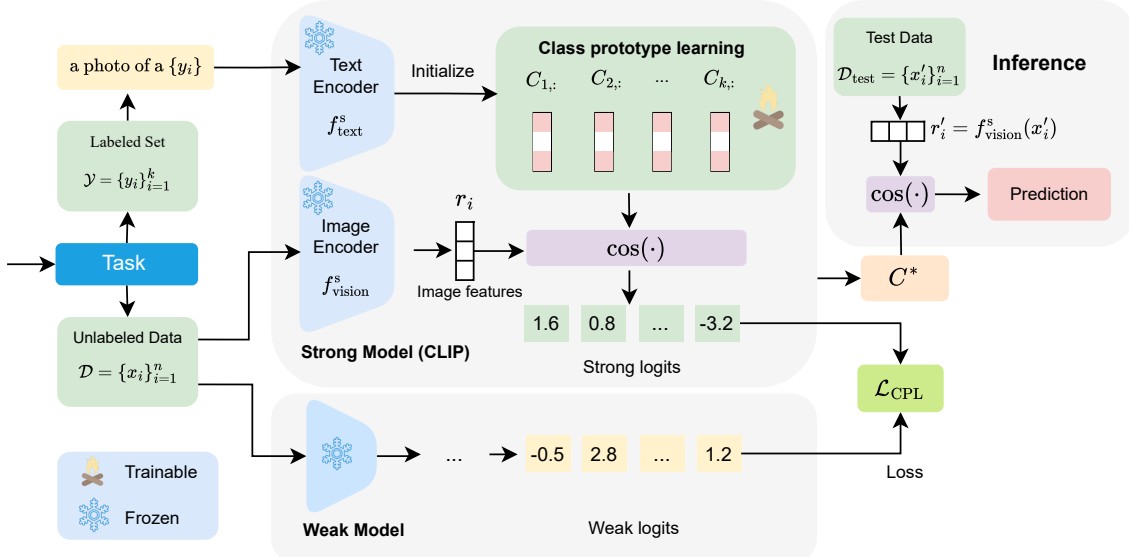

Figure 1: **Overview of the weak-to-strong process for enhancing strong model performance using weak model supervision**. Unlabeled data from a given task is fed into both a strong model (CLIP) and a weak model. The strong model uses an image encoder to generate image features ($\boldsymbol{r}_i$), which are compared with learnable class prototypes ($\boldsymbol{C}_{1,:}, \boldsymbol{C}_{2,:}, ..., \boldsymbol{C}_{k,:}$) through cosine similarity to produce strong logits. Concurrently, the weak model generates weak logits from the same data. Our alignment loss ($L_{\mathrm{CPL}}$ in Eq. 5) is computed between the strong logits (based on the prototype matrix $\boldsymbol{C}$) and weak logits. For test data, the image features ($\boldsymbol{r}'_i$) extracted from the strong model $f^{\mathrm{s}}$ are compared with the learned prototype matrix $\boldsymbol{C}^*$ to make predictions, aiming to improve the strong model's classification performance in the given task.

where $N$ is the batch size, $(\boldsymbol{i}_n, \boldsymbol{j}_n)$ denotes the index pairs of positive examples, and $\tau$ is a temperature parameter. This contrastive learning approach enables CLIP to achieve remarkable zero-shot classification performance across various tasks, leveraging its pretrained representations $\boldsymbol{z}_i$ and $\boldsymbol{z}_j$ without task-specific training.

**CLIP linear probs.** The standard method to fine-tune pre-trained VLMs, e.g., CLIP, involves training a linear classifier on the feature representations extracted from these pre-trained models. This approach mirrors how Radford et al. (2021) evaluated the transferability of CLIP, treating pre-trained models primarily as feature extractors. This method is generally more efficient because only the parameters of the additional classification heads need to be trained. The formula is:

$$\hat{\boldsymbol{p}} = \mathrm{softmax}(\boldsymbol{W} \cdot f^{\mathrm{s}}_{\mathrm{vision}}(\boldsymbol{x}) + b) \tag{1}$$

where $f^{\mathrm{s}}_{\mathrm{vision}}(\boldsymbol{x})$ denotes the feature representation extracted from the pre-trained CLIP model for an input $\boldsymbol{x}$, $\boldsymbol{W}$ represents the weights of the linear classifier, $b$ is the bias term, and $\hat{p}$ is the predicted probability distribution over the classes. The parameters $\boldsymbol{W}$ and $b$ are learned during the training process on the downstream task's labelled data. This approach leverages the rich feature representations learned by CLIP during its pre-training phase, enabling efficient and effective adaptation to new tasks with minimal additional training. In this method, $f^{\mathrm{s}}_{\mathrm{text}}$ is not used during this training process.

**CLIP prompt tuning.** A recent mainstream approach to more effectively adapt VLMs involves learning customized prompts (Zhou et al., 2022b). This method fine-tunes the input prompts that guide the model's attention and feature extraction processes. Mathematically, this approach can be represented as:

$$\hat{\boldsymbol{p}} = \cos(f^{\mathrm{s}}_{\mathrm{vision}}(\boldsymbol{x}), f^{\mathrm{s}}_{\mathrm{text}}(\{\boldsymbol{t}_i\}_{i=1}^k)) \tag{2}$$

where $f_{\text{vision}}^{\text{s}}(\boldsymbol{x})$ denotes the feature representation extracted from the vision encoder for an input $\boldsymbol{x}$, and $f_{\text{text}}^{\text{s}}(\{\boldsymbol{t}_i\}_{i=1}^k)$ denotes the feature representation extracted from the text encoder for a set of prompts $\{\boldsymbol{t}_i\}_{i=1}^k$, where $\boldsymbol{t}_i = \{\boldsymbol{v}_1, \boldsymbol{v}_2, ..., \boldsymbol{v}_k, \{\text{classname}_i\}\}$, with $\boldsymbol{v}_1, \boldsymbol{v}_2, ..., \boldsymbol{v}_k$ representing the learned prompt vectors and $\{\text{classname}_i\}$ being the target class name. The function cos represents the cosine similarity between the vision and text feature representations. The parameters of the prompt vectors $\boldsymbol{v}_1, \boldsymbol{v}_2, ..., \boldsymbol{v}_k$ are learned during the training process, enabling the model to better align the vision and language features for the specific downstream task. Unlike in the previous method, $f_{\text{text}}^{\text{s}}$ is utilized during the training.

## 4 Weak-to-Strong Learning for CLIP-based Classification

In this section, we first introduce the problem formulation and describe our proposed method CPL. Additionally, the overall procedure is shown in Figure 1, and the algorithm is shown in Algorithm 1.

**Problem formulation.** In this paper, we consider a scenario involving a weakly pre-trained model $f^{\text{w}}$ and a strongly pre-trained model $f^{\text{s}}$, where $f^{\text{s}}$ generally exhibits better generalization due to more parameters or extensive training data. In this paper, we consider $f^{\text{s}}$ to be a VLM model, e.g., CLIP (Radford et al., 2021), while $f^{\text{w}}$ is a vision model. Given a target task, we have a dataset consisting of $n$ unlabeled samples $\mathcal{D} = \{\boldsymbol{x}_i\}_{i=1}^n$, $m$ labeled test samples[1] $\mathcal{D}_{\text{test}} = \{(\boldsymbol{x}_i^{\text{te}}, \boldsymbol{y}_i)\}_{i=1}^m$ and a label set $\mathcal{Y} = \{\boldsymbol{y}_i\}_{i=1}^k$, where $k$ represents the number of category and each $\boldsymbol{y}_i$ represent one semantic label. We apply the weak model $f^{\text{w}}$ to $\mathcal{D}$ to generate predictions, which we refer to as *weak supervision*, represented by a weakly supervised dataset $\mathcal{D}_w = \{(\boldsymbol{x}_i, f^{\text{w}}(\boldsymbol{x}_i))\}_{i=1}^n$. The task of weak-to-strong generalization is to fine-tune the strong model $f^{\text{s}}$ with the weakly supervised dataset $\mathcal{D}_w$ to enhance its classification capabilities on the test dataset $\mathcal{D}_{\text{test}}$.

**Class prototype learning.** Empirical evidence (Figure 2a and 2b) indicates that previous VLM fine-tuning approaches, including linear probs and prompt tuning, applied in weak-to-strong generalization often result in strong models overfitting to the weak models. Consequently, this leads to the strong models performing close to the weak models on test sets. To address this, we aim to learn the set of class prototypes as a matrix $\boldsymbol{C} \in \mathbb{R}^{k \times d}$, where $k$ is the total number of classes and each row $\boldsymbol{C}_{i,:} \in \mathrm{R}^{1 \times d}$ is the class prototype for each class $i$ based on the feature embeddings of training images belonging to that class, which encapsulate the characteristics of each class. The prototype representation $\boldsymbol{C}_{i,:}$ for each class $i$ can be initialized by the text embedding corresponding to a textual description of the class label. For instance, $\boldsymbol{C}_{i,:}$ could be initialized with the text embedding of "a photo of a {label}" extracted by CLIP text encoder, where label represents the class label name.

When presented with an input image $\boldsymbol{x} \in \mathbb{R}^{h \times w \times c}$ in $\mathcal{D}$, we compute its visual feature embedding $f_{\text{vision}}^{\text{s}}(\boldsymbol{x})$, where $f_{\text{vision}}^{\text{s}}(\boldsymbol{x}) \in \mathbb{R}^{d \times 1}$. Then, the cosine similarity between the image embedding $f_{\text{vision}}^{\text{s}}(\boldsymbol{x})$ and each class prototype $\boldsymbol{C}_{i,:}$ is calculated as the logits $\boldsymbol{z}^{\text{s}}$. Mathematically, the unnormalized logit for $i$-th class (i.e., the $i$-th element in $\boldsymbol{z}$) regarding $\boldsymbol{x}$ is computed as:

$$z_i^{\text{s}}(\boldsymbol{C}_{i,:}, \boldsymbol{x}) = \frac{\boldsymbol{C}_{i,:} f_{\text{vision}}^{\text{s}}(\boldsymbol{x})}{\|\boldsymbol{C}_{i,:}\| \|f_{\text{vision}}^{\text{s}}(\boldsymbol{x})\|}, \tag{3}$$

where this cosine similarity operation measures the alignment between the image and class centroids, providing a measure of the image's association with each class. Subsequently, a softmax function can be applied to the logits to obtain class probabilities.

**Weak-to-strong alignment.** The ultimate aim of weak-to-strong alignment is to elicit the capabilities of a much stronger model using weak supervision from a weaker model (Burns et al., 2023). Unlike knowledge distillation, where the stronger model serves as the teacher and the weaker model as the student, weak-to-strong alignment reverses these roles. Here, the weaker model acts as the teacher guiding the stronger model. A straightforward approach to this challenge is to use knowledge distillation methods (Hinton et al., 2015) to make the strong model's behavior agree with that of the weak model. Most logit-based KD methods utilize the KL divergence, which quantifies the amount of information lost when approximating one probability

---

[1]This test set is only used for testing phrase.

---

**Algorithm 1:** Weak-to-strong Generalization for VLMs

---

**Input** : An unlabeled set: $\mathcal{D} = \{\boldsymbol{x}_i\}_{i=1}^n$; a test set: $\mathcal{D}_{\text{test}} = \{\boldsymbol{x}_i^{\text{te}}\}_{i=1}^m$; a label set: $\mathcal{Y} = \{\boldsymbol{y}_i\}_{i=1}^k$; a strong model: $f^{\text{s}}(\cdot)$; learnable class prototypes: $\boldsymbol{C} \in \mathbb{R}^{k \times d}$; maximum epochs: $T_{\max}$; alignment loss function: $\mathcal{L}_{\text{CPL}}(\cdot, \cdot)$.

**1: Obtain** $f^{\text{s}}_{\text{vision}}(\cdot)$, $f^{\text{s}}_{\text{text}}(\cdot)$ from $f^{\text{s}}(\cdot)$;
**2: Initialize** class prototypes $\boldsymbol{C}$ where $\boldsymbol{C}_{i,:} = f^{\text{s}}_{\text{text}}$("a photo of a $\{\boldsymbol{y}_i\}$");
**for** $T = 1$ **to** $T_{\max}$ **do**
  **3: Fetch** mini-batch $\mathcal{B}$ in $\mathcal{D}$;
  **4: Compute** the average loss $\mathcal{L} = \frac{1}{|\mathcal{B}|} \sum_{\boldsymbol{x} \in \mathcal{B}} \mathcal{L}_{\text{CPL}}(\boldsymbol{C}, \boldsymbol{x})$;
  **5: Update** class prototypes $\boldsymbol{C}$ using Adam (Kingma & Ba, 2014) and the average loss $\mathcal{L}$;
**end**
**6: Get** learned class prototypes $\boldsymbol{C}^*$;
**7: Obtain** test feature embeddings $\{\boldsymbol{r}_i\}_{i=1}^m = \{f^{\text{s}}_{\text{vision}}(\boldsymbol{x}^{\text{te}})\}_{\boldsymbol{x}^{\text{te}} \in \mathcal{D}_{\text{test}}}$;
**8: Compute** predicted logits $\{\boldsymbol{z}_i\}_{i=1}^m$ where $\boldsymbol{z}_i = \cos(\boldsymbol{C}^*, r_i)$;
**9: Compute** prediction $\hat{\mathcal{Y}} = \{\arg\max_j \boldsymbol{z}_{i,j}^{\text{s}}\}_{i=1}^n$;

**Output** : $\hat{\mathcal{Y}}$

---

distribution with another. Therefore, for each $\boldsymbol{x}$ in $\mathcal{D}$, given the logits of the weak model, $\boldsymbol{z}^{\text{w}}(\boldsymbol{x}) = f^{\text{w}}(\boldsymbol{x})$, and those of the strong model $\boldsymbol{z}^{\text{s}}$ (using Eq. 3), we convert them into the softened probability vector $\boldsymbol{p}^{\text{w}}$ and $\boldsymbol{p}^{\text{s}}$. The $i$-th value of $\boldsymbol{p}^{\text{w}}$ or $\boldsymbol{p}^{\text{s}}$ is computed by a *softmax* function with a temperature hyperparameter $\tau$, which is denoted by

$$p_i^{\text{w}}(\boldsymbol{x}) = \frac{\exp(z_i^{\text{w}}(\boldsymbol{x})/\tau)}{\sum_{j=1}^k \exp(z_j^{\text{w}}(\boldsymbol{x})/\tau)}, \quad p_i^{\text{s}}(\boldsymbol{C}_{i,:}, \boldsymbol{x}) = \frac{\exp(z_i^{\text{s}}(\boldsymbol{C}_{i,:}, \boldsymbol{x})/\tau)}{\sum_{j=1}^k \exp(z_j^{\text{s}}(\boldsymbol{C}_{j,:}, \boldsymbol{x})/\tau)}. \tag{4}$$

Thus, the loss value of each $\boldsymbol{x}$ in $\mathcal{D}$ is realized by minimizing the KL divergence between softened probability vectors of weak and strong models, which is defined as:

$$\mathcal{L}_{\text{CPL}}(\boldsymbol{C}, \boldsymbol{x}) = \text{KL}(\boldsymbol{p}^{\text{s}}(\boldsymbol{C}, \boldsymbol{x}) \parallel \boldsymbol{p}^{\text{w}}(\boldsymbol{x})) = \sum_{i=1}^k p_i^{\text{w}}(\boldsymbol{C}_{i,:}) \log \frac{p_i^{\text{w}}(\boldsymbol{C}_{i,:})}{p_i^{\text{s}}(\boldsymbol{C}_{i,:}, \boldsymbol{x})}. \tag{5}$$

We demonstrate the overall algorithm in Algorithm 1. The algorithm for weak-to-strong generalization in VLMs begins by initializing class prototypes using text embeddings from the strong model. During training, mini-batches of unlabeled data are processed to obtain feature embeddings and generate logits from both the strong and weak models. The alignment loss between these logits is computed to update the class prototypes iteratively. Once training is complete, the learned class prototypes are used to compute feature embeddings from the test data, generate logits through cosine similarity, and predict the labels by selecting the class with the highest logit value.

## 5 Experiments

In this section, we evaluate the performance of our method by a series of experiments and various ablation studies. The implementation details can be found in Appendix 5.

**Datasets.** In our exploration of weak-to-strong scenarios, we turn to the challenging and relatively large dataset: *DomainNet* (Peng et al., 2019). Comprising six diverse domains, each housing 345 categories of common objects, *DomainNet* offers a rich landscape for analysis. These domains encompass a range of visual styles and sources: Clipart, featuring a collection of clipart images; Infograph, presenting infographic images with specific objects; Painting, showcasing artistic renditions of objects in the form of paintings; Quickdraw, housing drawings from the popular game "Quick Draw!" by worldwide players; Real, encompassing photographs and real-world images; and Sketch, containing sketches of various objects. Refer to Table 2 for detailed statistics into each domain.

**Experimental setup.** In our experiments, each domain within the DomainNet dataset is treated as an individual task, resulting in a total of 6 tasks under consideration. To investigate weak-to-strong generalization within the setting of VLMs, we design these steps to simulate the problem:

(1) **Dataset splitting**: Referring to Table 2, each domain is divided into a training set $\mathcal{D}_{\text{train}}$ and a test set $\mathcal{D}_{\text{test}}$. The test set $\mathcal{D}_{\text{test}}$ is further partitioned into $\mathcal{D}_{\text{hold}}$ and $\mathcal{D}'_{\text{test}}$, comprising 80% and 20% of $\mathcal{D}_{\text{test}}$ respectively. (2) **Create the weak model**: The training data $\mathcal{D}_{\text{train}}$ is utilized to fine-tune the weak model, employing ground truth labels. Evaluation occurs in $\mathcal{D}_{\text{test}}$, termed as *weak performance*. (3) **Weak supervision set generation**: The *weak supervision* set $\mathcal{D}'_{\text{hold}}$ is generated by the weak model from $\mathcal{D}_{\text{hold}}$, replacing ground labels with logits produced by the weak model. (4) **Strong model training with weak supervisor**: Initially, $\mathcal{D}'_{\text{hold}}$ is split into 80% and 20% portions for strong model fine-tuning and parameter tuning, respectively. The strong model is then fine-tuned in the holdout training set. The final performance is assessed in $\mathcal{D}_{\text{test}}$, labeled as *weak-to-strong performance*. (5) **Strong model training with ground truth labels as ceiling**: Finally, the strong model undergoes fine-tuning on $\mathcal{D}_{\text{hold}}$ (with ground truth labels) to represent *strong ceiling performance*.

**Baselines.** In exploring the weak-to-strong problem within the VLM setting, we investigate different fine-tuning strategies. Initially, Radford et al. (2021) assessed CLIP's transferability via *linear probs* (LP) across many datasets. Subsequent research focused on *textual prompting* (TP) (Zhou et al., 2022b), where a learnable prompt is learned from a small target dataset. This method is data-efficient and demonstrates good generalization effects. Prompt tuning has emerged as a popular method for adapting VLMs to downstream tasks (Wu et al., 2023). Thus, we adopt linear probs and prompt tuning as our foundational fine-tuning strategies within the realm of weak-to-strong generalization. In addition, we compare our method with the following learning strategies:

(1) **Cross entropy** (CE): Utilized in studies by (Radford et al., 2021; Zhou et al., 2022b), cross-entropy measures the disparity between one-hot ground truth label distribution and model prediction probability. It serves as a straightforward baseline for this task. (2) **Knowledge distillation** (KD) (Hinton et al., 2015) transfer knowledge from a strong model to a smaller one, serving as a fundamental baseline due to its simplicity and effectiveness. (3) **Auxiliary confidence loss** (AuxConf) is proposed by Burns et al. (2023), which excels in balancing direct learning from the weak model with the inherent capacity of the strong model. (4) **Adaptive Confidence loss** (AdaptConf) is introduced by Guo et al. (2024) that dynamically adjusts weights based on confidence levels, enabling the strong model to discern when to prioritize its predictions or follow the guidance of the weak model.

**Implementation details.** In this section, we provide an overview of the implementation details regarding our proposed method and comparative baseline methods on simulation experiments. The code is mainly based on Pytorch and the Huggingface library. We employed ResNet and ViT as the weak model and CLIP as the strong model, for our task. The evaluation is performed in five random seeds. During training, we used a test batch size of 2048 for evaluation. The weak model was trained for 3 epochs with a batch size of 512 and a learning rate of 1e-3, whereas the strong model underwent 10 epochs with the same batch size and a learning rate of 1e-2. The learning rate was adjusted dynamically, and a warm-up ratio of 0.1 was utilized. We also ensured the loading of the best model at the end of training based on the validation set. All our experiments are conducted using a single A100 GPU with 40GB of memory, supported by 8 CPU workers and 64GB of RAM.

**Experiment results.** The results presented in Table 1 provide a comprehensive evaluation of various methods across multiple domains within the DomainNet dataset. Each method's efficacy is assessed based on its accuracy in six distinct domains: Clipart, Infograph, Painting, Quickdraw, Real, and Sketch. Notably, our proposed method (CPL) exhibits remarkable performance, achieving the highest average accuracy of 64.74%. This signifies a substantial improvement over baseline methods, with CPL outperforming the best-performing baseline by notable margins, showcasing gains of 2.41%, 10.11%, 3.19%, -0.19%, 1.41%, and 3.67% across respective domains.

Table 1: **Performance on DomainNet datasets across different methods and styles**. This table showcases the results in accuracy (%) of various methods on different styles within the DomainNet datasets, including Clipart, Infograph, Painting, Quickdraw, Real, and Sketch. The average performance across all styles is also listed. The compared methods include CE+LP, KD+LP, AuxConf+LP, AdaptConf+LP, CE+TP, KD+TP, AuxConf+TP, and AdaptConf+TP, with CPL yielding the highest performance in most categories. The final row, Δ, represents the improvement margin of CPL over other methods. CPL is used as the strong ceiling performance, which is the best among the LP, TP, and CPL

| Method | DomainNet | | | | | | Avg. |
|---|---|---|---|---|---|---|---|
| | Clipart | Infograph | Painting | Quickdraw | Real | Sketch | |
| Weak | 67.15 | 31.71 | 67.90 | 46.70 | 85.10 | 52.26 | - |
| Strong Ceiling | 74.27 | 50.84 | 72.24 | 49.92 | 85.34 | 66.64 | - |
| CE+LP | 66.97 | 30.55 | 64.90 | 45.59 | 82.69 | 55.28 | 57.66 |
| KD+LP | 70.69 | 35.78 | 68.28 | **48.15** | 83.66 | 59.88 | 61.07 |
| AuxConf+LP | 67.41 | 18.37 | 66.92 | 30.59 | 84.00 | 56.79 | 54.01 |
| AdaptConf+LP | 70.68 | 33.78 | 67.86 | 47.80 | 83.84 | 60.18 | 60.69 |
| CE+TP | 62.02 | 29.25 | 62.68 | 44.82 | 81.16 | 52.51 | 55.41 |
| KD+TP | 69.57 | 36.03 | 68.37 | 47.34 | 83.70 | 59.93 | 60.82 |
| AuxConf+TP | 69.20 | 20.97 | 68.61 | 43.60 | 83.92 | 58.49 | 57.47 |
| AdaptConf+TP | 69.97 | 35.96 | 68.18 | 47.42 | 83.59 | 59.95 | 60.85 |
| CPL (Ours) | **73.10** | **46.14** | **71.80** | 47.96 | **85.41** | **64.01** | **64.74** |
| Δ | 2.41 | 10.11 | 3.19 | -0.19 | 1.41 | 3.83 | 3.67 |

In the domain of QuickDraw, it is evident that CLIP demonstrates a lower zero-shot ability, suggesting significant disparities between the data distribution in QuickDraw and the CLIP training data. This observation underscores the challenge of generalizing CLIP to the QuickDraw domain effectively. Surprisingly, in such cases, the straightforward KD approach emerges as the most effective method, outperforming more sophisticated techniques. This phenomenon suggests that the inherent structure of the KD method enables it to leverage available information optimally, leading to superior performance despite the substantial dissimilarities between the CLIP and QuickDraw domains.

In the context of the Infograph domain, the weak model exhibits notably inferior performance compared to all other domains. Conversely, our proposed method demonstrates the most substantial performance improvement, showcasing a significant gain in accuracy as compared to both the weak model and other competing methods. This highlights the effectiveness of our approach in addressing the challenges specific to the Infograph domain, where the weak model struggles to generalize effectively. The considerable performance gain achieved by our method underscores its ability to capture and leverage domain-specific features, resulting in improved accuracy and robustness in handling Infograph data.

As shown in Table 4, our BLIP-based method (CLIP (CPL)) consistently outperforms all baselines across the DomainNet dataset. It achieves the highest average accuracy, with noticeable improvements over strong adaptation methods such as AdaptConf+TP and AuxConf+TP. These results demonstrate the effectiveness of our approach in leveraging BLIP representations for improved domain generalization.

**Ablation on different CLIP variants.** Table 5 compares our method against a wide range of baselines using different CLIP backbones (ViT-B/16 and ViT-L/16) on the Clipart domain. Our approach achieves the best performance across both variants. These results confirm the effectiveness and robustness of our method under both low- and high-capacity model settings, outperforming baseline methods.

**Ablation on OfficeHome.** As shown in Table 6, our method achieves the best performance across all four domains of the OfficeHome dataset, outperforming existing baselines by a clear margin. Specifically, it attains an average accuracy of **75.36%**, improving over the best prior method (AdaptConf+TP) by **2.16%**. The consistent gains across domains—Art (**+2.22%**), Clipart (**+2.20%**), Product (**+1.67%**), and Real-World (**+2.57%**)—highlight the robustness and generalization ability of our approach.

Table 2: **DomainNet statistics.** This table provides statistics for the DomainNet dataset (Peng et al., 2019) across different styles: Clipart, Infograph, Painting, Quickdraw, Real, and Sketch. It includes the number of classes (#Classes), the number of training samples (#Train), the number of test samples (#Test), and the total number of samples (#Total) for each style. Each style has 345 classes, with varying numbers of training and test samples.

|  | Clipart | Infograph | Painting | Quickdraw | Real | Sketch |
|---|---|---|---|---|---|---|
| #Classes | 345 | 345 | 345 | 345 | 345 | 345 |
| #Train | 33525 | 36023 | 50416 | 120750 | 120906 | 48212 |
| #Test | 14604 | 15582 | 21850 | 51750 | 52041 | 20916 |
| #Total | 48,129 | 51,605 | 72,266 | 172,500 | 172,947 | 69,128 |

Table 3: **Performance Comparison of Different Weak Models**. This table presents the results in accuracy (%) of various methods applied to weak models, including Resnet-18, Resnet-26, Resnet-34, Cvt-13, and Convnext-tiny-224. The average performance across all models is also provided. The strong ceiling performance is given for reference. The methods compared are CE+LP, KD+LP, AuxConf+LP, AdaptConf+LP, CE+TP, KD+TP, AuxConf+TP, and AdaptConf+TP, with CPL showing the best performance. The final row, $\Delta$, indicates the improvement margin of CPL over other methods.

| Method | Weak Models | | | | | Avg. |
|---|---|---|---|---|---|---|
|  | Resnet-18 | Resnet-26 | Resnet-34 | Cvt-13 | Convnext-tiny-224 |  |
| Weak | 55.22 | 57.2 | 59.96 | 51.33 | 69.47 | - |
| Strong Ceiling |  |  | 74.27 |  |  | - |
| CE+LP | 64.28 | 64.19 | 64.72 | 62.19 | 69.5 | 64.98 |
| KD+LP | 66.80 | 67.26 | 68.53 | 65.89 | 71.53 | 68.00 |
| AuxConf+LP | 66.19 | 67.02 | 66.21 | 62.52 | 71.10 | 66.61 |
| AdaptConf+LP | 67.72 | 68.13 | 68.83 | 66.95 | 71.42 | 68.61 |
| CE+TP | 62.22 | 63.35 | 64.04 | 61.02 | 67.71 | 63.67 |
| KD+TP | 65.71 | 66.43 | 67.34 | 64.50 | 69.31 | 66.66 |
| AuxConf+TP | 66.39 | 67.04 | 67.47 | 66.04 | 69.56 | 67.30 |
| AdaptConf+TP | 65.91 | 66.69 | 67.42 | 65.31 | 69.04 | 66.87 |
| CPL (BLIP) | **72.25** | **71.84** | **72.06** | **71.91** | **72.47** | **72.11** |
| $\Delta$ | 4.53 | 3.71 | 3.23 | 4.96 | 0.94 | 3.50 |

**Ablation on different weak supervision.** Table 3 illustrates our approach to various forms of weak supervision across different models in detail, such as Resnet models (He et al., 2016) (Resnet-18, Resnet-26, Resnet-34), Cvt-13 (Wu et al., 2021), and Convnext-tiny-224 (Liu et al., 2022). The experiment was conducted on the DomainNet Clipart domain, revealing a diverse range of performances from different weak models, with accuracy scores spanning from 55.2% to 69.47%. Notably, our method consistently achieved the best weak-to-strong generalization performance among all the weak models tested, closely approximating the strong ceiling performance, which was benchmarked at 74.27%.

Our method's superior performance is evident across various weak supervision techniques. The results show that while other methods improved performance to varying degrees, none matched the consistency and high performance of our method. For instance, our approach significantly outperformed the baseline weak supervision models, achieving top accuracy scores such as 72.25% for Resnet-18, 71.84% for Resnet-26, 72.06% for Resnet-34, 71.91% for Cvt-13, and 72.47% for Convnext-tiny-224. On average, our method achieved a performance gain of 3.5%, underscoring its superior ability to enhance model accuracy through improved weak supervision techniques.

The performance gains highlight the incremental improvements our method brings compared to other approaches. These improvements range from 0.94% to 4.96%, demonstrating our method's ability to consistently push model performance closer to the strong ceiling benchmark. From this table, it is evident

Table 4: **Performance on DomainNet datasets across different methods and styles**. This table showcases the results in accuracy (%) of various methods on different styles within the DomainNet datasets, including Clipart, Infograph, Painting, Quickdraw, Real, and Sketch. The average performance across all styles is also listed. The compared methods include CE+LP, KD+LP, AuxConf+LP, AdaptConf+LP, CE+TP, KD+TP, AuxConf+TP, and AdaptConf+TP, with CPL yielding the highest performance in most categories. The final row, Δ, represents the improvement margin of CPL over other methods. CPL is used as a strong ceiling performance, which is the best among the LP, TP, and CPL.

| Method | DomainNet | | | | | | Avg. |
| --- | --- | --- | --- | --- | --- | --- | --- |
| | Clipart | Infograph | Painting | Quickdraw | Real | Sketch | |
| Weak | 60.23 | 28.45 | 63.12 | 42.98 | 80.56 | 50.34 | - |
| Strong Ceiling | 75.12 | 52.78 | 73.65 | 50.89 | 86.12 | 67.45 | - |
| CE+LP | 61.78 | 27.69 | 61.34 | 40.21 | 81.23 | 52.10 | 54.06 |
| KD+LP | 68.12 | 33.47 | 65.89 | 46.21 | 83.02 | 57.45 | 59.36 |
| AuxConf+LP | 65.89 | 19.45 | 63.78 | 32.10 | 82.34 | 54.67 | 53.37 |
| AdaptConf+LP | 69.23 | 31.89 | 66.14 | 44.75 | 83.47 | 58.12 | 58.93 |
| CE+TP | 59.67 | 27.12 | 60.45 | 39.88 | 79.89 | 50.76 | 52.96 |
| KD+TP | 67.34 | 34.23 | 66.10 | 45.12 | 82.78 | 58.23 | 59.30 |
| AuxConf+TP | 66.78 | 21.10 | 65.23 | 41.89 | 82.90 | 55.89 | 55.63 |
| AdaptConf+TP | 67.89 | 33.12 | 65.45 | 44.12 | 83.10 | 57.67 | 58.56 |
| CPL (Ours) | **72.45** | **45.23** | **70.12** | **47.89** | **85.23** | **63.45** | **64.06** |
| Δ | 3.22 | 11.00 | 3.98 | 1.68 | 1.76 | 5.22 | 3.84 |

Table 5: **Comparison across CLIP variants (ViT-B/16 and ViT-L/16) on the Clipart domain**. Accuracy (%) is reported for each method. Our method achieves the best performance across both variants, confirming its robustness.

| Backbone | Weak | Strong Ceiling | CE+LP | KD+LP | AuxConf+LP | AdaptConf+LP | CE+TP | KD+TP | AuxConf+TP | AdaptConf+TP | CPL (Ours) |
| --- | --- | --- | --- | --- | --- | --- | --- | --- | --- | --- | --- |
| ViT-B/16 | 57.45 | 76.89 | 66.23 | 67.89 | 69.45 | 71.12 | 68.56 | 70.12 | 72.34 | 73.78 | **75.12** |
| ViT-L/16 | 60.78 | 80.12 | 69.34 | 71.12 | 73.23 | 75.34 | 72.45 | 74.23 | 76.45 | 77.89 | **79.45** |

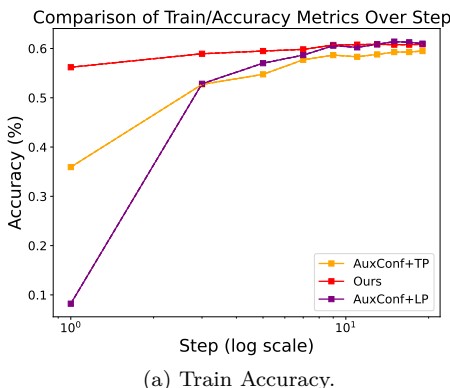

(a) Train Accuracy.

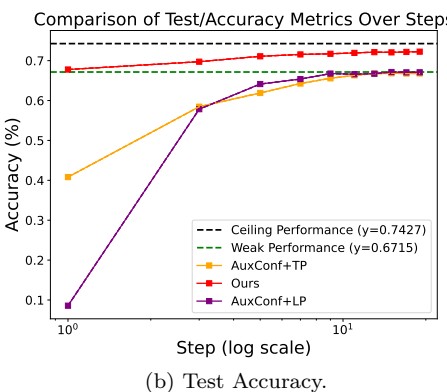

(b) Test Accuracy.

Figure 2: **Comparison of train and test accuracy metrics over training steps**. It shows the comparison of train (a) and test (b) accuracy metrics for different methods over training steps. Methods include AuxConf+TP, Ours, and AuxConf+LP. "Ours" demonstrates the highest accuracy, nearing the ceiling performance ($y = 0.7427$) and surpassing weak performance ($y = 0.6715$) in both the training and testing phases.

that weak-to-strong generalization is feasible in the VLMs setting. By utilizing supervision from weak models, our strong model has attained results that are close to the ceiling performance.

Table 6: **Performance on OfficeHome domains (Art, Clipart, Product, Real-World)**. Accuracy (%) is reported for each domain and averaged across them. Our method (**CLIP (CPL)**) achieves the best results across all domains.

| Method | Art | Clipart | Product | Real-World | Avg. |
|---|---|---|---|---|---|
| Weak | 54.78 | 50.45 | 63.12 | 66.95 | - |
| Strong Ceiling | 79.10 | 65.34 | 85.01 | 87.23 | - |
| CE+LP | 62.10 | 60.12 | 72.05 | 76.90 | 67.79 |
| KD+LP | 63.80 | 61.55 | 74.45 | 78.22 | 69.00 |
| AuxConf+LP | 65.23 | 61.90 | 75.78 | 79.90 | 70.20 |
| AdaptConf+LP | 66.89 | 62.78 | 77.01 | 81.12 | 71.45 |
| CE+TP | 64.89 | 61.45 | 75.56 | 79.05 | 70.11 |
| KD+TP | 66.23 | 62.56 | 76.45 | 80.56 | 71.02 |
| AuxConf+TP | 67.89 | 64.12 | 77.89 | 81.78 | 72.42 |
| AdaptConf+TP | 68.23 | 64.78 | 78.67 | 83.10 | 73.20 |
| **CLIP (CPL)** | **70.45** | **66.98** | **80.34** | **85.67** | **75.36** |
| $\Delta$ | 2.22 | 2.20 | 1.67 | 2.57 | 2.16 |

Table 7: **Average Performance Comparison of Different Tuning Methods in Accutacy (%)**.

| Method | Performance |
|---|---|
| Text Encoder | 70.34 |
| $C$ | 74.42 |

**Ablation on different tuning methods.** We have conducted an ablation study to compare the performance of tuning C versus tuning the text encoder. The results have been shown in Table 7. Our findings indicate that tuning $C$ yields better performance than tuning the text encoder. The study by Wu et al. (2023) shows that prompt tuning for VLMs is more robust to noisy labels compared to fine-tuning.

**Analysis of our method.** In Figures 2a and 2a, we demonstrate the training and test accuracy over training steps for our method compared to two baseline methods, AuxConf+TP and AuxConf+LP. The training accuracy plot shows that all methods eventually converge to an accuracy of around 0.6. Specifically, our method shows a rapid and consistent improvement, achieving high training accuracy more quickly than the other methods. The baseline methods, AuxConf+TP and AuxConf+LP, also improve but at different rates, with AuxConf+TP showing a steadier progression and AuxConf+LP catching up later in the process. In the test accuracy plot, the differences between the methods become more pronounced. While AuxConf+TP and AuxConf+LP exhibit similar weak performance levels, struggling to surpass a certain threshold, our method showcases significantly better performance. It not only achieves higher test accuracy but also maintains this performance consistently over the steps, closely approaching the ceiling performance.

## 6 Limitation

Since the core problem we aim to address in this research has not yet emerged, we currently lack access to superintelligence models. Although our experiments rely on simulations, these simulated scenarios do not fully replicate the complexities of the actual challenge we anticipate. Consequently, there exists a significant gap between our simulated experiments and the real-world problem. This discrepancy implies that the methods demonstrating success in our current simulations may not necessarily prove effective when applied to the final real-world task. Therefore, while our current research provides valuable insights and progress, it remains crucial to acknowledge these limitations and continue refining our approaches to better align with the ultimate goal of weak-to-strong alignment.

# 7 Conclusion

In conclusion, traditional alignment techniques for LLMs, which rely heavily on human supervision, such as RLHF, face significant challenges due to the intricacy of model outputs and the inefficiency of requiring substantial human feedback. To address this, a novel approach has recently been proposed where a weaker model supervises a much stronger one. Extending this concept to VLMs leverages these insights, adapting the proven benefits to a multi-modal context. Hence, we introduced a method called CPL, which effectively enhances the classification capabilities of VLMs with weak supervision. Our simulation experiments validate the effectiveness of this weak-to-strong approach. Extensive experimental results demonstrate that our method significantly improves performance across various benchmarks. These results underscore the potential of weak supervision as a powerful tool in the alignment, offering a promising avenue for future research and application.

# Acknowledgments

This research was supported by The University of Melbourne's Research Computing Services and the Petascale Campus Initiative. FL is supported by the Australian Research Council (ARC) with the grant number DE240101089, LP240100101, DP230101540 and the NSF&CSIRO Responsible AI program with grant number 2303037.

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
