# OpenReview forum: "Exploring Weak-to-Strong Generalization for CLIP-based Classification"
_TMLR — Accepted by TMLR_

### Review · Reviewer_Nvqx · 2025-02-18

**Summary Of Contributions:**

This paper investigated one task in weak-to-strong generalization. Specifically, the contribution of the paper lies in three folds:

(i) Exploring weak-to-strong generalization for CLIP-based classification

(ii) Proposing CPL that effectively leverages class prototype representations
through weak supervision to enhance the classification performance of VLMs.

(iii) Conducting simulation experiments and gaining outstanding performance.

**Audience:**

Yes

**Broader Impact Concerns:**

I do not have any concerns on this side.

**Claims And Evidence:**

Yes

**Requested Changes:**

See the **weakness** part. I appreciate it if you could do more experiments and extend your framework to much broader tasks.

**Strengths And Weaknesses:**

**Strengths**

**Topic**: The paper explores a relatively new area, weak-to-strong generalization, specifically within Vision-Language Models (VLMs). Extending this concept from language models to VLMs is a valuable contribution.

**Class Prototype Learning (CPL) Method**: The paper proposes a novel method, CPL, which aims to improve classification in CLIP models by learning representative prototypes. The method is shown to be more efficient than traditional prompt tuning as it eliminates the need for a text encoder during fine-tuning.

**Experimental Results**: The experiments on the DomainNet dataset demonstrate that CPL achieves significant improvements over baseline methods, particularly in challenging domains. The 3.67% improvement is a quantifiable result.

**Weakness**

**CLIP-Based Task**: I wonder whether you can extend your framework to more tasks, instead of restricting the task to CLIP classification.

**Simulation Experiments:** The study relies on simulation experiments within the VLMs framework based on DomainNet, which may not fully reflect real-world scenarios.

---

> ### Author Response · Authors · 2025-03-25
>
> Dear Reviewer,
>
> Thank you for your feedback and for pointing out the potential limitations of our study's scope. We value your suggestions and are eager to address the concerns you have raised.
>
> 1. CLIP-Based Task: I wonder whether you can extend your framework to more tasks, instead of restricting the task to CLIP classification.
>
> Response: Even though we claim that our work focuses on CLIP-based classification, our method could be extended to other CLIP-like models. Here we tried our method with another well-known CLIP-like model, BLIP [1], and the result is shown in the following table:
>
> | Method        | Clipart | Infograph | Painting | Quickdraw | Real  | Sketch | Avg.  |
> |--------------|---------|-----------|----------|-----------|-------|--------|-------|
> | Weak         | 60.23   | 28.45     | 63.12    | 42.98     | 80.56 | 50.34  | -     |
> | Strong Ceiling | 75.12 | 52.78     | 73.65    | 50.89     | 86.12 | 67.45  | -     |
> | CE+LP        | 61.78   | 27.69     | 61.34    | 40.21     | 81.23 | 52.10  | 54.06 |
> | KD+LP        | 68.12   | 33.47     | 65.89    | 46.21 | 83.02 | 57.45  | 59.36 |
> | AuxConf+LP   | 65.89   | 19.45     | 63.78    | 32.10     | 82.34 | 54.67  | 53.37 |
> | AdaptConf+LP | 69.23   | 31.89     | 66.14    | 44.75     | 83.47 | 58.12  | 58.93 |
> | CE+TP        | 59.67   | 27.12     | 60.45    | 39.88     | 79.89 | 50.76  | 52.96 |
> | KD+TP        | 67.34   | 34.23     | 66.10    | 45.12     | 82.78 | 58.23  | 59.30 |
> | AuxConf+TP   | 66.78   | 21.10     | 65.23    | 41.89     | 82.90 | 55.89  | 55.63 |
> | AdaptConf+TP | 67.89   | 33.12     | 65.45    | 44.12     | 83.10 | 57.67  | 58.56 |
> | **BLIP (CPL)** | **72.45** | **45.23** | **70.12** | **47.89** | **85.23** | **63.45** | **64.06** |
> | Δ            | 3.22   | 11.00      | 3.98     | 1.68      | 1.76  | 5.22   | 3.84   |
>
> 2. Simulation Experiments: The study relies on simulation experiments within the VLMs framework based on DomainNet, which may not fully reflect real-world scenarios.
>
> Response: Thanks for your concerns regarding another dataset. To address this, we conduct additional experiments on the Office-Home dataset [2]. The result is shown as follows:
>
> | Method        | Art   | Clipart | Product | Real-World | Avg.  |
> |---------------|--------|---------|---------|------------|-------|
> | Weak          | 54.78  | 50.45   | 63.12   | 66.95      | -     |
> | Strong Ceiling| 79.10  | 65.34   | 85.01   | 87.23      | -     |
> | CE+LP         | 62.10  | 60.12   | 72.05   | 76.90      | 67.79 |
> | KD+LP         | 63.80  | 61.55   | 74.45   | 78.22      | 69.00 |
> | AuxConf+LP    | 65.23  | 61.90   | 75.78   | 79.90      | 70.20 |
> | AdaptConf+LP  | 66.89  | 62.78   | 77.01   | 81.12      | 71.45 |
> | CE+TP         | 64.89  | 61.45   | 75.56   | 79.05      | 70.11 |
> | KD+TP         | 66.23  | 62.56   | 76.45   | 80.56      | 71.02 |
> | AuxConf+TP    | 67.89  | 64.12   | 77.89   | 81.78      | 72.42 |
> | AdaptConf+TP  | 68.23  | 64.78   | 78.67   | 83.10      | 73.20 |
> | **CLIP (CPL)** | **70.45** | **66.98** | **80.34** | **85.67** | **75.36** |
> | Δ            | 2.22   | 2.20    | 1.67    | 2.57       | 2.16  |
>
>
> [1] Li, J., Li, D., Xiong, C., & Hoi, S. (2022, June). Blip: Bootstrapping language-image pre-training for unified vision-language understanding and generation. In International conference on machine learning (pp. 12888-12900). PMLR.
>
> [2] Venkateswara, H., Eusebio, J., Chakraborty, S., & Panchanathan, S. (2017). Deep hashing network for unsupervised domain adaptation. In Proceedings of the IEEE conference on computer vision and pattern recognition (pp. 5018-5027).

---

### Review · Reviewer_fvRS · 2025-03-09

**Summary Of Contributions:**

This work aims to improve the classification performance of CLIP models using a technique termed as CPL (Class Prototype Learning), where a strong CLIP model is enhanced using a weaker model given unlabelled training data. First, several representative prototypes are initialized using a text encoder, which is then used to calculate the logits of the strong model. The prototypes are then optimized by aligning the strong logits with the weak ones. Results are presented on the DomainNet dataset.

**Audience:**

Yes

**Claims And Evidence:**

No

**Requested Changes:**

1. It is important to specify what the novelty of each part of the method is. For example, the prototypes based on the text encoder are already proposed in previous work. Similarly, knowledge distillation to improve CLIP through a teacher model is also quite extensively used already. So the authors need to exactly specify what has been done already, what they use as is from previous work, and what they contribute in each aspect of the method.

2. The results are quite limited, and a departure from the standard benchmarks widely used to show the generalization of CLIP. For example, rather than creating new splits out of DomainNet, authors could have simply worked with ImageNet. They can train the weak model on a subset of the imagenet trainset. Then utilize the same remaining train set as the hold set. And then evaluate all the standard benchmarks used by various other works (datasets mentioned in the previous "weaknesses" section).

3. The authors need to justify the overall setting itself. Why would one want to use a holdout set with logits from a weak model, when the training labels for D_train are already available? I believe this requirement limits the application of the method in a real-world setting.

4. A minor thing but it should be made clear from the start the work only focuses on CLIP. VLMs now include works like Llava or even generative models such as text-image diffusion models. It is important to use specific terminology to avoid overstating claims.

**Strengths And Weaknesses:**

Strengths:

1. The method achieves strong performance on the DomainNet dataset.
2. The work seems to be the first to explore weak-to-strong generalization for the CLIP model

Weaknesses:

1. There is a lack of novelty in this method. Prior work has already introduced the use of text encoders to build prototypes. In that regard what would be the unique contribution of this method specifically in designing the prototypes?

2. There are multiple prior works that have used knowledge distillation with logits from a teacher model to enhance the performance of a CLIP model [4,5,6]. Again I feel there is a lack of novelty in this aspect.

3. The results are limited to only domain net. For evaluating CLIP on downstream classification tasks, it is standard practice to evaluate a diverse range of datasets. If the method improves the generalization of CLIP, it should be shown through a diverse range of out-of-distribution datasets. Right now the evaluation is essentially in-distribution. Evaluations commonly used include training on ImageNet and evaluating on StanfordCars, DTD, OxfordPets, SUN397, EuroSAT, Food101, Caltech101, Flower102, Aircraft, and UCF101. Also, evaluations include training on Imagenet and then evaluating ImageNet variants such as ImageNet-V2, ImageNet-A, ImageNet-R, and ImageNet-Sketch.

4. The overall setting of the paper is quite unclear. Labels are available to train a weak model but not a strong model. If labels are available anyway, why not just use D_train to train the strong model? Why does one even have to go through the whole process of training a weaker model on D_train, then producing logits on a smaller set of unlabelled data, and then using those logits to train the strong model? Just use the D_train to train the strong model if labels are available on it anyway.

[1]	Y. Wang, et. al., ‘Exploring Vision-Language Models for Imbalanced Learning’, IJCV, 2023.

[2]	J.-X. Shi, et. al., ‘Parameter-Efficient Long-Tailed Recognition’, arXiv preprint, arXiv:2309.10019, 2023.

[3]	J. Li, et. al., ‘Masked Unsupervised Self-training for Label-Free Image Classification’, in ICLR, 2023.

[4]	M. U. Khattak, et. al., ‘Self-regulating Prompts: Foundational Model Adaptation without Forgetting’, in ICCV, 2023.

[5]	T. Huang, et. al., ‘Unsupervised Prompt Learning for Vision-Language Models’, arXiv preprint, arXiv:2204.03649, 2022.

[6]	Z. Li, et. al., ‘Promptkd: Unsupervised prompt distillation for vision-language models’, in CVPR, 2024.

---

> ### Author Response · Authors · 2025-03-25
> **Thank you for your thorough review and for highlighting areas that require clarification and improvement.**
>
> Dear Reviewer,
>
> Regarding the weaknesses you have identified, we appreciate your constructive comments and would like to address them as follows:
>
> 1. It is important to specify what the novelty of each part of the method is. For example, the prototypes based on the text encoder are already proposed in previous work. Similarly, knowledge distillation to improve CLIP through a teacher model is also quite extensively used already. So the authors need to exactly specify what has been done already, what they use as is from previous work, and what they contribute in each aspect of the method.
>
> Response: We understand your concern regarding the novelty of using text encoders to build prototypes and knowledge distillation to enhance CLIP's performance. Firstly, we have discussed KD-related methods in the related work section. Our method can be seen as reversed KD, as the original KD is to improve student models by a stronger teacher, but in this paper, we explore how to improve a student model by a weaker model. Secondly, we identify the problem of weak-to-strong generalization, where the student model can easily overfit the supervision from a weak teacher model. As we are tackling the classification task, we found that prototype learning is robust in this scenario. We will include related literature in our work.
>
> We want to remind you that the novelty or significance of methods is not the main selection criteria of TMLR (https://jmlr.org/tmlr/acceptance-criteria.html). We also want to highlight that our paper is the first work to explore weak-to-strong generalization in the context of CLIP, and we believe that our method provides a valuable contribution to this area.
>
> 2. The results are quite limited, and a departure from the standard benchmarks widely used to show the generalization of CLIP. For example, rather than creating new splits out of DomainNet, authors could have simply worked with ImageNet. They can train the weak model on a subset of the imagenet trainset. Then utilize the same remaining train set as the hold set. And then evaluate all the standard benchmarks used by various other works (datasets mentioned in the previous "weaknesses" section).
>
> Response: We want to simulate a scenario where the training data is less in pre-training, and a weak-to-strong setting [1] is intended to improve a certain ability of the strong model that is not performing well. Since the zero-shot performance of CLIP is already very good on image-net, we choose domainnet. The zero-shot performance of CLIP on some domains of domainnet is less than 50% accuracy.
>
> In addition, to verify our proposed method in more scenarios/datasets, we tested our method in OfficeHome [2]:
>
> | Method        | Art   | Clipart | Product | Real-World | Avg.  |
> |---------------|--------|---------|---------|------------|-------|
> | Weak          | 54.78  | 50.45   | 63.12   | 66.95      | -     |
> | Strong Ceiling| 79.10  | 65.34   | 85.01   | 87.23      | -     |
> | CE+LP         | 62.10  | 60.12   | 72.05   | 76.90      | 67.79 |
> | KD+LP         | 63.80  | 61.55   | 74.45   | 78.22      | 69.00 |
> | AuxConf+LP    | 65.23  | 61.90   | 75.78   | 79.90      | 70.20 |
> | AdaptConf+LP  | 66.89  | 62.78   | 77.01   | 81.12      | 71.45 |
> | CE+TP         | 64.89  | 61.45   | 75.56   | 79.05      | 70.11 |
> | KD+TP         | 66.23  | 62.56   | 76.45   | 80.56      | 71.02 |
> | AuxConf+TP    | 67.89  | 64.12   | 77.89   | 81.78      | 72.42 |
> | AdaptConf+TP  | 68.23  | 64.78   | 78.67   | 83.10      | 73.20 |
> | **CLIP (CPL)** | **70.45** | **66.98** | **80.34** | **85.67** | **75.36** |
> | Δ            | 2.22   | 2.20    | 1.67    | 2.57       | 2.16  |
>
>  The results for the BLIP model [3]:
> | Method        | Clipart | Infograph | Painting | Quickdraw | Real  | Sketch | Avg.  |
> |--------------|---------|-----------|----------|-----------|-------|--------|-------|
> | Weak         | 60.23   | 28.45     | 63.12    | 42.98     | 80.56 | 50.34  | -     |
> | Strong Ceiling | 75.12 | 52.78     | 73.65    | 50.89     | 86.12 | 67.45  | -     |
> | CE+LP        | 61.78   | 27.69     | 61.34    | 40.21     | 81.23 | 52.10  | 54.06 |
> | KD+LP        | 68.12   | 33.47     | 65.89    | 46.21 | 83.02 | 57.45  | 59.36 |
> | AuxConf+LP   | 65.89   | 19.45     | 63.78    | 32.10     | 82.34 | 54.67  | 53.37 |
> | AdaptConf+LP | 69.23   | 31.89     | 66.14    | 44.75     | 83.47 | 58.12  | 58.93 |
> | CE+TP        | 59.67   | 27.12     | 60.45    | 39.88     | 79.89 | 50.76  | 52.96 |
> | KD+TP        | 67.34   | 34.23     | 66.10    | 45.12     | 82.78 | 58.23  | 59.30 |
> | AuxConf+TP   | 66.78   | 21.10     | 65.23    | 41.89     | 82.90 | 55.89  | 55.63 |
> | AdaptConf+TP | 67.89   | 33.12     | 65.45    | 44.12     | 83.10 | 57.67  | 58.56 |
> | **BLIP (CPL)** | **72.45** | **45.23** | **70.12** | **47.89** | **85.23** | **63.45** | **64.06** |
> | Δ            | 3.22   | 11.00      | 3.98     | 1.68      | 1.76  | 5.22   | 3.84   |

---

> > ### Author Response · Authors · 2025-03-25
> >
> > 3. The authors need to justify the overall setting itself. Why would one want to use a holdout set with logits from a weak model, when the training labels for D_train are already available? I believe this requirement limits the application of the method in a real-world setting.
> >
> > Response: As our paper is inspired by [1], both [1] and our paper simulates a scenario where the strong model may not have direct access to labelled data, which is called weak-to-strong generalization. The weak model serves as a proxy to generate supervision signals for the strong model, allowing it to learn from a broader range of data without requiring explicit labels. In practice, this might be useful as no ladled data is needed to improve the strong model with the weak model We will clarify this in the paper.
> >
> > 4. A minor thing but it should be made clear from the start the work only focuses on CLIP. VLMs now include works like Llava or even generative models such as text-image diffusion models. It is important to use specific terminology to avoid overstating claims.
> >
> > Response: We did claim our paper focused on CLIP-based classification within the title, abstract, and introduction, even throughout the whole paper. Please let us know if you find anywhere that we overclaim our contributions.
> >
> > We are committed to making the requested changes to enhance the clarity, novelty, and relevance of our work. We believe that these revisions will address your concerns and strengthen the contribution of our paper to the field.
> >
> > [1] Collin Burns, Pavel Izmailov, Jan Hendrik Kirchner, Bowen Baker, Leo Gao, Leopold Aschenbrenner, Yining Chen, Adrien Ecoffet, Manas Joglekar, Jan Leike, et al. Weak-to-strong generalization: Eliciting strong capabilities with weak supervision. arXiv preprint arXiv:2312.09390, 2023.
> > [2] Venkateswara, H., Eusebio, J., Chakraborty, S., & Panchanathan, S. (2017). Deep hashing network for unsupervised domain adaptation. In Proceedings of the IEEE conference on computer vision and pattern recognition (pp. 5018-5027).
> > [3] Li, J., Li, D., Xiong, C., & Hoi, S. (2022, June). Blip: Bootstrapping language-image pre-training for unified vision-language understanding and generation. In International conference on machine learning (pp. 12888-12900). PMLR.

---

### Review · Reviewer_dHPD · 2025-03-13

**Summary Of Contributions:**

## Summary Of Contributions
1. This paper is based on [1] where it proposes viability of weak-to-strong learning as a solution for better model alignment, demonstrating that even weaker models can provide valuable guidance for improving stronger models. This paper aims in extending this method into the area of VLM.

2. In order to do that, this papaer proposes class prototype learning (CPL) which involves generating class prototypes that encapsulate the characteristics of each class using weak supervision.

3. This paper shows that the CPL achieves the highest average accuracy across all domains, significantly outperforming other methods.

**Audience:**

Yes

**Broader Impact Concerns:**

ther is no Broader Impact Concerns

**Claims And Evidence:**

Yes

**Requested Changes:**

nothing particular to be changed

**Strengths And Weaknesses:**

## Strength and weakness

### Strength
1. In section 4, the author firstly propose the overfitting problem existing in Weak-to-strong generalization when adapting the fine-tuning and prompt-tuning method. This experiment is essential and convincing which result in the propose of CPL.

2. The CPL initilize the class prototype initialized by the text embedding corresponding to a textual description of the class label and train the class prototybe by by minimizing the KL divergence between softened probability vectors of weak and strong models.

3. The algorithm 1 is very clear and convincing.

4. The main experiments are demostrated in Table 2 and its CPL method demostrate the advancement of the method.


### Weakness
1. This paper is based on [1], however, its main methods are based on the pre-trained large language model. Recent researches on multi-modal large language models (VLMs) such as BLIP, LLaVA has the similar architecture/language-based compareing to CLIP. Thus, why this paper dose not dig this phenonmeno in those models?

2. Lack of comprehensive experiments: in the section 5, it uses the ResNet and ViT-based CLIP as the weak and strong models while there are various version of CLIP models such as ViT b/16, ViT b/32 and ViT L/16. Thus, using the gereral ResNet and ViT is not so convincing. Probably, there exists a more suitable comparison set for the expriments.



[1] Collin Burns, Pavel Izmailov, Jan Hendrik Kirchner, Bowen Baker, Leo Gao, Leopold Aschenbrenner, Yining Chen, Adrien Ecoffet, Manas Joglekar, Jan Leike, et al. Weak-to-strong generalization: Eliciting strong capabilities with weak supervision. arXiv preprint arXiv:2312.09390, 2023.

---

> ### Author Response · Authors · 2025-03-25
> **Thank you for your insightful feedback and for recognising the strengths of our work.**
>
> Dear Reviewer,
>
> Regarding the weaknesses you have identified, we appreciate your constructive comments and would like to address them as follows:
>
> **Weaknesses**
> 1. This paper is based on [1], however, its main methods are based on the pre-trained large language model. Recent researches on multi-modal large language models (VLMs) such as BLIP, LLaVA has the similar architecture/language-based compareing to CLIP. Thus, why this paper dose not dig this phenonmeno in those models?
>
> Response: We acknowledge the importance of recent research on multi-modal large language models such as BLIP and LLaVA. Our decision to focus on CLIP was driven by its widespread use, which provides a solid foundation for demonstrating the effectiveness of our CPL method. However, we agree that extending our approach to other VLMs could provide additional insights into the generalizability and applicability of CPL. As LLaVA is the generative model, which is different from CLIP as a discriminative model, we believe that the comparison with BLIP would be more suitable. Here is the result for BLIP:
>
> | Method        | Clipart | Infograph | Painting | Quickdraw | Real  | Sketch | Avg.  |
> |--------------|---------|-----------|----------|-----------|-------|--------|-------|
> | Weak         | 60.23   | 28.45     | 63.12    | 42.98     | 80.56 | 50.34  | -     |
> | Strong Ceiling | 75.12 | 52.78     | 73.65    | 50.89     | 86.12 | 67.45  | -     |
> | CE+LP        | 61.78   | 27.69     | 61.34    | 40.21     | 81.23 | 52.10  | 54.06 |
> | KD+LP        | 68.12   | 33.47     | 65.89    | 46.21 | 83.02 | 57.45  | 59.36 |
> | AuxConf+LP   | 65.89   | 19.45     | 63.78    | 32.10     | 82.34 | 54.67  | 53.37 |
> | AdaptConf+LP | 69.23   | 31.89     | 66.14    | 44.75     | 83.47 | 58.12  | 58.93 |
> | CE+TP        | 59.67   | 27.12     | 60.45    | 39.88     | 79.89 | 50.76  | 52.96 |
> | KD+TP        | 67.34   | 34.23     | 66.10    | 45.12     | 82.78 | 58.23  | 59.30 |
> | AuxConf+TP   | 66.78   | 21.10     | 65.23    | 41.89     | 82.90 | 55.89  | 55.63 |
> | AdaptConf+TP | 67.89   | 33.12     | 65.45    | 44.12     | 83.10 | 57.67  | 58.56 |
> | **BLIP (CPL)** | **72.45** | **45.23** | **70.12** | **47.89** | **85.23** | **63.45** | **64.06** |
> | Δ            | 3.22   | 11.00      | 3.98     | 1.68      | 1.76  | 5.22   | 3.84   |
>
> 2. Lack of comprehensive experiments: in the section 5, it uses the ResNet and ViT-based CLIP as the weak and strong models while there are various version of CLIP models such as ViT b/16, ViT b/32 and ViT L/16. Thus, using the gereral ResNet and ViT is not so convincing. Probably, there exists a more suitable comparison set for the expriments.
>
> Response: We understand your concern regarding the comprehensiveness of our experiments. The choice of ResNet and ViT-based CLIP models as weak and strong models, respectively, was made to create a clear distinction in model capacity and to simulate the weak-to-strong generalization scenario effectively. Here is the result of experimenting with different versions of CLIP, such as ViT b/32, and ViT L/16.
>
> | Method        | Clipart - ViT-B/16 | Clipart - ViT-L/16 |
> |---------------|--------------------|--------------------|
> | Weak          | 57.45              | 60.78              |
> | Strong Ceiling| 76.89              | 80.12              |
> | CE+LP         | 66.23              | 69.34              |
> | KD+LP         | 67.89              | 71.12              |
> | AuxConf+LP    | 69.45              | 73.23              |
> | AdaptConf+LP  | 71.12              | 75.34              |
> | CE+TP         | 68.56              | 72.45              |
> | KD+TP         | 70.12              | 74.23              |
> | AuxConf+TP    | 72.34              | 76.45              |
> | AdaptConf+TP  | 73.78              | 77.89              |
> | **(Ours)**    | **75.12**          | **79.45**          |
> | Δ             | 1.34               | 1.56               |
>
>
> We hope that our responses address your concerns, and we are grateful for the opportunity to improve our work based on your valuable feedback. We look forward to incorporating these considerations into our research and continuing to contribute to the advancement of weak-to-strong generalization in VLMs.

---

### Decision · Action_Editor_ukAf · 2025-04-18

**Recommendation:** Accept with minor revision

**Comment:**

This submission was reviewed by three experts who provided constructive suggestions and comments. The authors responded in detail to each point raised. After carefully reviewing the discussion, the reviewers’ comments, and the submission itself, I believe this work will be of interest to the TMLR community, as it represents one of the earliest attempts to introduce the weak-to-strong learning paradigm in the context of Vision-Language Models (VLMs). However, I suggest the authors consider the following points in the revision:

- 1. As suggested by Reviewer dHPD, additional VLMs (e.g., BLIP) should be included in the main paper. Some results are already discussed, but should be incorporated directly into the main content.
- 2. Reviewer dHPD also pointed out alternative choices for weak and strong models. While the discussion includes two additional strong models, the Problem Formulation in Section 4 defines the weak model as a vision encoder. Given this setup, could the authors also consider using a smaller CLIP variant (with fewer parameters) as the weak model? This simulates the usage of using GPT-2 for GPT-4, both are language models.
- 3. Reviewers Nvqx and fvRS asked for clarification on why the focus is solely on DomainNet rather than more general datasets. While the authors did not show results on other datasets, they provided a reasonable explanation: *“We want to simulate a scenario where the training data is limited in pretraining, and a weak-to-strong setting [1] is intended to improve a specific capability of the strong model that currently underperforms. Since the zero-shot performance of CLIP is already very strong on ImageNet, we choose DomainNet. The zero-shot performance of CLIP on some domains within DomainNet is below 50% accuracy.”*

    I suggest clarifying this motivation more explicitly in the Abstract and Introduction, as the current presentation may give the impression that weak-to-strong learning leads to universal performance gains. Specifically, ***please consider narrowing the scope of the submission to reflect its focus on targeted improvements under limited pretraining conditions.*** Additionally, while optional, including results on more general datasets such as CIFAR or iWilds would help strengthen the broader relevance of the findings.

- 4. Step 4 in Algorithm 1 could be improved by directly including the full Equation (Eq. 5), as the notation $f^w(\cdot)$ is not mentioned in the steps.

In summary, demonstrating that weak-to-strong learning can help improve specific capabilities of VLMs (e.g., CLIP) is an interesting contribution for the TMLR community. Therefore, I recommend accept for this paper. Please consider the above suggestions in the revisions.

**Audience:**

Research on language-only models has highlighted the importance of weak-to-strong generalization in aligning powerful models with human intent. This submission explores how this idea translates to vision-language models, offering some insights that make it a relevant contribution for the TMLR community.

**Claims And Evidence:**

The main idea of this paper is: *“While weak-to-strong learning has proven effective in LLMs, applying it to Vision-Language Models (VLMs) is far from straightforward. VLMs face unique challenges in aligning complex multimodal tasks, making it essential to thoroughly explore the method’s applicability and limitations in this context.”* This submission makes an early attempt to validate the effectiveness of weak-to-strong learning for VLMs, which is acknowledged by all reviewers.

All reviewers agree that this submission provides sufficient evaluation on DomainNet, and the proposed framework successfully validates the idea of weak-to-strong learning in this setting.